# The Impact of the Establishment of the Mount Wuyi National Park on the Livelihood of Farmers

Zhen Yang [1], Jinjie Ren [2] and Dahong Zhang [1,*]

1   School of Economics and Management, Beijing Forestry University, Beijing 100083, China;
    yz20080808@bjfu.edu.cn
2   School of Landscape Architecture, Beijing Forestry University, Beijing 100083, China; renjinjie98@bjfu.edu.cn
*   Correspondence: zdh@bjfu.edu.cn

**Abstract:** The construction of national parks has a profound impact on the production and life of surrounding farmers. Based on the survey data of 354 farmers around the Mount Wuyi National Park, this paper uses the method of constructing a difference-in-difference model to analyze the impact of the Mount Wuyi National Park construction on the livelihood choices of surrounding farmers and the main income of families. In addition, this article analyzes the heterogeneity of surrounding farmers based on differences in tea planting area and farmers' income levels. The results show the following: First, compared with pre-establishment, farmers around the Mount Wuyi National Park still prefer tea-based agricultural employment or part-time employment. Second, after the establishment of the Mount Wuyi National Park, the main income of surrounding farmers' families is still agricultural income. Therefore, the authors of this article believe that it is necessary to further strengthen the protection of the livelihoods of surrounding farmers and moderately create more livelihood choices for surrounding farmers in order to achieve both ecological and economic benefits.

**Keywords:** Mount Wuyi National Park; farmers' livelihoods; main household income; difference-in-difference model





## 1. Introduction

The establishment of national parks is playing an increasingly important role in further improving China's ecological civilization construction. In 2013, the Third Plenary Session of the 18th Central Committee of the CPC defined the national park system, marking the entrance of the country into a stage of a protected area management system with the "national park system as the main body". In 2016, China established the first group of pilot national parks. In October 2021, Mount Wuyi, Amur Tiger and Amur Leopard, Sanjiangyuan, and the Giant Panda and Hainan Tropical Rainforest National Park were officially established, indicating that China's national park construction has made further progress in the community [1]. With the establishment of national parks, the impact on the production and life of surrounding farmers has gradually become a research hotspot [2]. As a typical area with relatively dense population within the national park, Mount Wuyi National Park officially became the first grouping of national park pilots in 2016. In 2017, Mount Wuyi National Park Administration was officially established. In 2021, Mount Wuyi National Park, as one of the first national parks, was officially established [3]. China's national parks, represented by the Mount Wuyi National Park, have been established in succession. In addition to the primary goal of ecological protection, ensuring the production and livelihood of farmers living around national parks and balancing the contrast between human and land have gradually become important goals for the establishment of national parks [4].

There have been numerous research results on the impact of farmers' livelihoods by scholars both domestically and internationally. Farmers' livelihoods are a closely related part of their production and life. In terms of employment, farmers' livelihoods mainly

include their participation in agricultural production activities, part-time employment, and non-agricultural employment [5]. Zhou et al. [6] believe that due to the acceleration of urbanization and industrialization in the country, the agricultural population continues to decline, making the non-agricultural employment of farmers an important foundation for the new urbanization strategy. Therefore, non-agricultural employment has the characteristics of being contemporary, long-term, and stable. Dong and Dong [7] also analyzed the problem of farmers' livelihood diversification. Research shows that the process of farmers' livelihood diversification can be summarized into four steps from the perspective of space transformation: living area transformation, social identity transformation, life meaning transformation and survival realm transformation [7]. From the perspective of the diversification of farmers' livelihoods and the transformation of farmers' identities, the new generation of farmers faces two career stages of "survival" and "development" in the process of urbanization. In the "survival" stage, employment is the most important problem faced by the new generation of farmers, and in the "development" stage, career growth becomes a new goal for the new generation of farmers within the process of urbanization [8]. In the application of models for the diversification of farmers' livelihoods, some scholars have analyzed and clarified the heterogenic and non-heterogenic characteristics of farmers' groups within the process of urbanization, as well as obstacles such as passive marginalization, active marginalization, and dual marginalization [9]. In addition, scholars have also focused on the influencing factors of the agricultural operating income of new vocational farmers under the diversification of farmers' livelihoods from a micro-individual perspective [10]. Thus, using farmers' livelihood diversification as the intermediary variable, the impact of agricultural mechanization, agricultural product prices, infrastructure construction, registered residence system, land management scale and other factors on farmers' income or urban-rural gap are explored in this in-depth study [11].

From the perspective of the theme of this article, there has been much research in the academic community on the impact of the establishment of national parks on the livelihoods of surrounding farmers. Firstly, in terms of the analysis of the relationship between national parks and farmers' livelihoods, some scholars believe that national parks and farmers' livelihoods have a coordinated and mutually beneficial relationship, which is common in most developed countries and a few developing countries [12]. A different viewpoint emphasizes a relationship of conflict between national park construction and farmers' livelihoods, which is mainly manifested as follows: the closer a farmer lives to a national park, the more singular their livelihood system will be, ultimately leading to a more tense relationship between the national park and farmers [13]. Wu and Wu believe that the establishment of some national parks does not have a significant impact on the livelihoods of farmers in remote provinces, mainly due to the relatively backward infrastructure in the province [14]. With the deepening of research, the academic community has also obtained new research results on the relationship between national parks and farmers' livelihoods: some scholars have found that the construction of Mount Wuyi National Park in Fujian had a dual impact on the livelihood of surrounding farmers: that is, ecological protection policies had a negative effect on farmers, while development guidance policies and ecological compensation policies had a positive effect on farmers' livelihoods. Overall, the construction of national parks has a greater positive impact on surrounding farmers [15]. At the same time, there is also research on the impact of national park construction on the local tourism industry, as well as the status, role, types of participation, and internal and external influencing factors of residents regarding tourism development [16]. In addition, some scholars used the explicit value evaluation method to analyze the degree of public participation in the construction of Qianjiangyuan National Park and then analyzed the impact on farmers' livelihoods [17]. Finally, regarding the impact of the external environment on farmers' livelihoods, scholars' analyses mainly included the use of time series analysis methods to construct econometric models, analyzing the impact of land acquisition for the construction of national parks in Vietnam on local environmental and ecological compensation policies, and then analyzing the impact on

local farmers' livelihoods [18]. Ensemble models have also been used to analyze the impact of natural disasters in Serbian national parks on the livelihoods of local farmers [19]. In summary, there are few current research results that systematically analyze the impact of national park construction on the livelihood choices of local farmers from a micro perspective, taking national park construction as the policy background. In addition, there is a lack of relevant literature analyzing the heterogeneity of farmers' livelihood choices from the perspectives of changes in the main composition of farmers' income and changes in non-agricultural income.

Based on the systematic analysis of the impact of the establishment of the Mount Wuyi National Park on the livelihood choice of local farmers and on the main composition of their income, this paper uses the construction of a difference-in-difference model ("DID model") to empirically test the above problems based on the survey data of 354 farmers in Mount Wuyi City [20] and analyzes the degree of difference in farmers' livelihood choices from the perspective of the main composition of farmers' income, changes in non-agricultural income, etc. The key questions to be answered in this paper are: First, will the establishment of the Mount Wuyi National Park significantly affect the livelihood choices of surrounding farmers? Second, will the establishment of the Mount Wuyi National Park change the main income structure of surrounding farmers? Thirdly, what impact will the changes in farmers' main income have on the non-agricultural income of farmers living around national parks?

Compared to previous studies, the marginal contribution of this article is mainly reflected in the following aspects: Firstly, data from micro-surveys were collected to analyze the impact of national parks on the livelihood choices of surrounding farmers. Secondly, when analyzing the main sources of income of farmers' families, the impact of national park construction on the income structure of the surrounding farmers was identified. Thirdly, the heterogeneity of the impact of national park construction on farmers' livelihood choices from the perspectives of farmers' occupational status and household income was analyzed and corresponding countermeasures and suggestions for the local government are proposed to further improve the management policies of national parks, optimize farmers' livelihood choices, and increase economic benefits.

## 2. Materials and Methods

### 2.1. Theoretical Analysis and Assumptions

In recent years, with the formal establishment and improvement of natural reserves, such as national parks, a number of laws, regulations, and rules such as the National Park Management Law (Trial) have also been introduced. With the increasing improvement of national park management, there are more policy guarantees for the protection of surrounding areas and support for farmers in the affected areas. On the one hand, these policies to some extent constrain the behavior of residents around national parks, promoting the continuous optimization of the national park environment. On the other hand, this has also led to more restrictions on the livelihood activities of farmers around the national park, thereby affecting their livelihood choices [21]. Based on the above theoretical analysis, this article proposes the following hypothesis 1.

**Hypothesis 1 (H1).** *With the construction of the Mount Wuyi National Park, there will be some restrictions on the livelihood choices of surrounding farmers, that is, they will still pay more attention to agriculture.*

According to the analysis of farmers' livelihood theory, farmers' incomes mainly consist of the following four parts. First is wage income, which refers to rural residents receiving labor remuneration through employment by others or selling their own labor. Second is operating income, where rural residents engage in production or business activities on a household basis to obtain income. Third is property income, which refers to the income obtained by the owner of financial assets or tangible non-productive assets providing funds or assets to other units for their disposal through returns. Fourthly, fiscal

income refers to the income obtained by rural households through government-provided transfer payments but does not include income generated by the free provision of fixed capital [22,23]. With the current operation of urbanization, industrialization, and ecological civilization construction activities, the livelihood of farmers is mainly affected by the following factors. Firstly, the continuous transfer of labor force, i.e., there is a high transfer level of rural labor force from agricultural employment to non-agricultural employment. Secondly, a large amount of agricultural land, especially arable land, has shifted from agricultural use to non-agricultural use. These changes, in turn, affect the main income of farmers' households [24]. In fact, scholars have conducted extensive research on the impact of non-agricultural income on farmers and have analyzed many influencing factors. The main entry point is the analysis of the influencing factors of the agricultural operating income of new vocational farmers under the diversification of farmers' livelihoods from a micro individual perspective [25]. In their research, scholars believe that, on the one hand, due to the impact of land loss and other factors, the improvement in farmers' employment quality is conducive to enhancing their urban adaptability, increasing farmers' urbanization capital, and ultimately contributing to the development quality of new urbanization, resulting in non-agricultural land, farmers' non-agricultural employment, and farmers' urbanization, and driving agricultural employment, non-agricultural employment, and part-time employment, thereby achieving an increase in farmers' income [26]. On the other hand, some scholars have argued from the perspective of spatial spillover of labor transfer that rural labor transfer has a role in narrowing the urban–rural income gap between individual farms and neighboring cities [27]. Based on the above theoretical analysis, this article proposes Hypothesis 2 accordingly.

**Hypothesis 2 (H2).** *With the construction of the Mount Wuyi National Park, the main income of surrounding farmers will change to non-agricultural income.*

To sum up, this paper establishes a theoretical analysis framework of the impact of the construction of the Mount Wuyi National Park on livelihood choices and the main incomes of surrounding farmers, as shown in Figure 1.

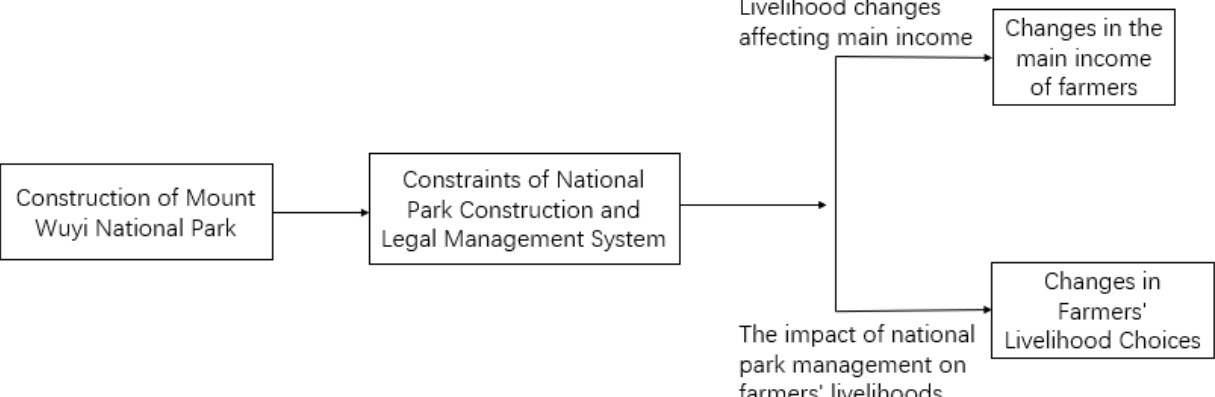

**Figure 1.** Impact of the Mount Wuyi National Park Construction on farmers' livelihood choices and main income.

### 2.2. Case Points and Data Sources

The Mount Wuyi National Park is located in the northwest of the Fujian Province and the southeast of the Jiangxi Province. It mainly encompasses Mount Wuyi City, Jianyang District, Shaowu City, Guangze County (Fujian area), and Yanshan County (Jiangxi area). The climate conditions are good, with abundant precipitation and abundant wildlife resources. Therefore, it is also known as the "paradise of snakes" and the "kingdom of birds". In 2016, the Mount Wuyi National Park was selected as the first grouping of national park pilots. In 2017, the Mount Wuyi National Park Administration was

officially established. On 30 September 2021, the Mount Wuyi National Park was officially established with the approval of the State Council. On October 12 of the same year, the Mount Wuyi National Park, the Sanjiangyuan National Park, the Giant Panda National Park, the Amur Tiger and Amur Leopard National Park, and the Hainan Tropical Rainforest National Park were included in the first group of national parks. In January 2022, the overall plan for the construction of the Mount Wuyi National Park was successfully passed. At the beginning of 2022, the Mount Wuyi Municipal Government formulated the Work Plan for the Construction of Protection and Development Belt around the Mount Wuyi National Park, encompassing 9 townships, 73 villages, and 113,900 people, including Xingcun Town, Wufu Town, Xingtian Town, and Yangzhuang Township. According to the plan, Mount Wuyi City will focus on strengthening the protection and inheritance of excellent traditional culture; comprehensively improve the infrastructure around the Mount Wuyi National Park; promote the comprehensive rural revitalization of villages surrounding the national parks; and promote the overall construction of tourism infrastructure around Mount Wuyi, thus striving to create a national cultural park. In terms of project planning, Mount Wuyi City plans to construct 49 projects with a total investment of CNY 22.947 billion. With the increasingly perfect planning and construction of Mount Wuyi National Park, the government's help regarding farmers' livelihoods mainly includes the following aspects. First, in order to support the development of the Wuyishan tea industry, the construction of the Wuyishan Tea Industrial Park has started. Second, the infrastructure construction of the Mount Wuyi Zhuzi Cultural Park and the promotion of local folk culture is being carried out. Third, the 251 km local eco-tourism loop project and the improvement of tourism service facilities in the Mount Wuyi Resort will be promoted in an integrated way. Fourthly, compensation for land acquisition from farmers for infrastructure construction will occur. Therefore, this paper takes the Mount Wuyi National Park as the research object, as it is representative.

In order to ensure the effectiveness of this study, the research team carried out a survey of the Mount Wuyi National Park in February 2023, mainly in the villages around the Mount Wuyi National Park within the jurisdiction of Mount Wuyi City. For the convenience of creating a comparison, the research group selected villages within and outside the scope of the national park for on-site sampling surveys. Of these, the surveyed farmers within the national park are the experimental group (large sample), while the surveyed farmers outside the national park are the control group (small sample). Ten villages within and outside the Mount Wuyi National Park were randomly selected as the research objects in this survey. The number of surveys required was set according to the actual population of each village. Finally, 360 farmers from 10 villages in 3 townships were surveyed through a total of 360 questionnaires, yielding 354 valid questionnaires with an effective questionnaire rate of 98.33%. Of these, there were 231 questionnaires from within the national park area and 123 questionnaires from outside the national park area. All the results of the survey were obtained through one-on-one interviews between the interviewer and the farmers. In order to highlight the differences in survey data, the research team obtained the farmers' personal information, family information, income and expenditure information, land use information, information on livelihood choices, and information on major changes in household incomes in 2015 (before the pilot of the Mount Wuyi National Park) and 2022 (after the official establishment of the Mount Wuyi National Park).

### 2.3. Variable Selection

First is the dependent variable. The explanatory variables in this study are the changes in farmers' livelihood choices and the changes in their main incomes. Some current studies point out that the implementation of certain ecological protection policies will affect farmers' livelihood choices [28]. At the same time, changes in farmers' livelihood methods will further affect changes in the main income of households [29]. Based on this, the research group selected one variable as the measure of farmers' livelihood choices, namely "whether the industry engaged by farmers' households is agriculture". In addition, when measuring

the main income of farmers, the research group chose "the main source of household income for farmers" as the measurement variable. This is used to estimate and analyze the impact of the construction of the Mount Wuyi National Park on farmers' livelihood choices and their main incomes. The choice of explanatory variables in this article is mainly based on the research topic, the impact of national park establishment on farmers' livelihoods, and the main income of households.

Second is the core explanatory variable. The core explanatory variable of this study is the interaction between the virtual variable of farmer groupings and the virtual variable of construction time before and after the establishment of the Mount Wuyi National Park. For the dummy variable group, a value of 1 was assigned to the surveyed farmers within the scope of the national park, otherwise a value of 0 was assigned. For the time dummy variable, the year 2022 (after the establishment of the Mount Wuyi National Park) was assigned a value of 1 and the year 2015 was assigned a value of 0. The selection of the core explanatory variables in this article was mainly based on the grouping of policy audiences and the virtual variables of policy time grouping.

Third are the control variables. This study controlled for variables that affect farmers' livelihood choices and their main incomes in regression analysis. Some existing research results shows that individual variables and family variables can affect farmers' livelihood choices [30]. Land use variables, attitude variables, and village level dummy variables can also affect the main income and livelihood choices of farmers to a certain extent [31]. Based on the above analysis, the specific control variables of this article are as follows. First is the personal characteristic variable of the household head, including whether they are village leaders or party members. The research results of some studies show that working in agriculture and other industries, as well as the experiences of party members and leaders, can provide more social capital to farmers, thereby promoting the adjustment of their livelihood methods. Secondly, household variables include the number of family members, labor force, and annual household savings. These variables mainly reflect the endowment conditions and material constraints within a family. Thirdly, land use variables include tea plantation area, bamboo area, annual government subsidies, and annual tea business income. The basis for setting these variables is that farmers around the Mount Wuyi National Park mainly plant tea and bamboo. According to field research, there is a positive relationship between the area of tea plantations and moso bamboo and overall total household income, while the area of cultivated land is generally inversely related to total household income. Fourthly, attitude variables include family attitudes towards career adjustment and the support of enterprises for farmers' employment. These external factors and variables can also have an impact on farmers' livelihood choices to a certain extent. Fifthly, village-level dummy variables are used to control the differences in the implementation of construction policies within and outside the scope of national parks, considering that the scope of national park involves different villages, and both complete and partial villages fall within this scope. The selection of control variables in this article is mainly based on the reference of relevant literature variable settings and the variables related to the research question.

### 2.4. Research Methods

First is the double difference model. In order to effectively evaluate the exogenous impact of national park establishment, this study uses the DID model to analyze the impact of the Mount Wuyi National Park construction on farmers' livelihood choices and their main incomes. In addition, the addition of certain covariates can strengthen the control of differences between the treatment group and the control group [32]. The DID model mainly analyzes the impact of a certain policy on local residents. Therefore, the double difference method selected in this article is targets the issues discussed in this article. The specific model of this study is as follows:

$$Option_{it} = a_0 + a_1 treated_i + a_2 post_i + a_3 treated_i \times post_i + a_4 X_{it} + \varepsilon_{it} \tag{1}$$

$$Source_{it} = \beta_0 + \beta_1 treated_i + \beta_2 post_i + \beta_3 treated_i \times post_i + \beta_4 X_{it} + \varepsilon'_{it} \tag{2}$$

In the above two formulas, $Option_{it}$, $Source_{it}$ are the explained variables, which, respectively, represent the impact of construction within and outside the Mount Wuyi National Park on farmers' livelihood choices and their main incomes; $i$ represents the farmer and $t$ represents the year; and $treated_i$ represents the dummy variable of the household grouping. If the household belongs to the scope of the national park, it is the treatment group, and the corresponding $treated_i = 1$, otherwise it is 0. $post_i$ represents the time dummy variable before and after the construction of the national park, with post before the construction as $post_i = 0$ and post after construction as $post_i = 1$. $treated_i \times post_i$ represents the core explanatory variable, which is the interaction between the dummy variable of the farmer grouping and the dummy variable of the establishment time of the national park. $X_{it}$ is the control variable. $\varepsilon_{it}$, $\varepsilon'_{it}$ represents a random disturbance term that varies with individuals and time. $a_0$, $\beta_0$ represents a constant term, and $a_1$, $a_2$, $a_3$, $a_4$, $\beta_1$, $\beta_2$, $\beta_3$, $\beta_4$ represent the estimated coefficients of the explanatory variable.

Second is GIS mapping technology.

The research group obtained the corresponding vector data through discussion with the Mount Wuyi National Park Administration. And using GIS mapping technology, the research team carried out a survey of the location of 10 villages in the Mount Wuyi National Park. The specific location is shown in Figure 2 [33]. This article uses GIS mapping technology to reproduce the villages interviewed by the research team during the research process. This demonstrates the scientific and rational selection of research areas.

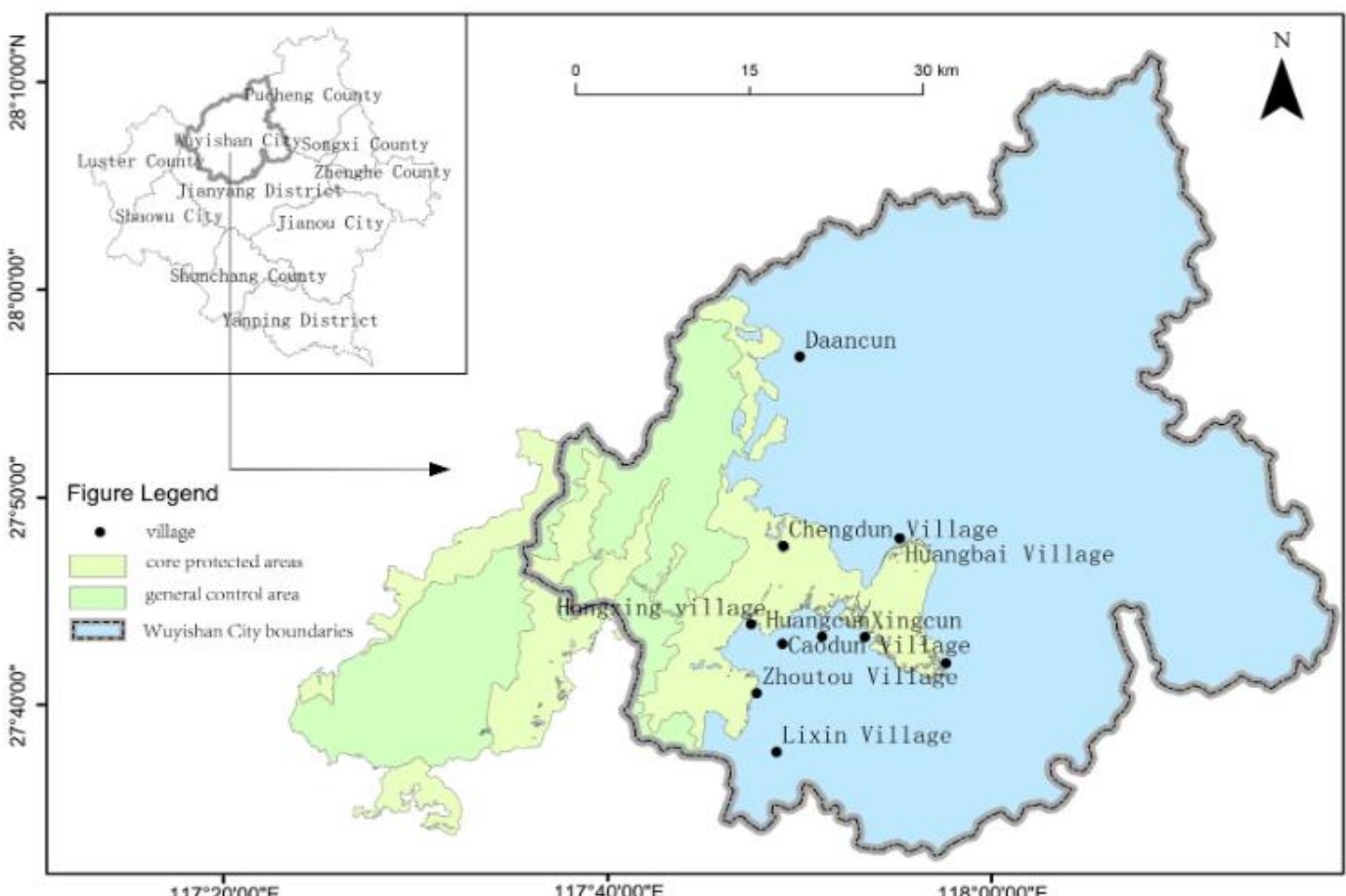

**Figure 2.** Schematic diagram of villages investigated in the Mount Wuyi National Park.

## 3. Results

### 3.1. Descriptive Statistics

According to Table 1, descriptive statistics were obtained regarding farmers' livelihood choices and their main incomes. The results show that the livelihood choices of farmers around the Mount Wuyi National Park in agriculture decreased from 0.47 in 2015 to 0.20 in 2022 and 0.10 for both years of non-agricultural employment. The variable of part-time jobs increased from 0.43 in 2015 to 0.70 in 2022. The variable of the main source of income increased from 2.00 in 2015 to 2.19 in 2022. Compared to 2015, in 2022, except for a decrease in the variables of agricultural work hours, family members, labor force, and cultivated land area, all other covariates showed varying degrees of increase. According to the relevant literature, an important aspect of the impact on farmers' incomes after the transformation of their livelihoods is the change to the main source of income [34]. The results of the descriptive statistical analysis preliminarily support the judgment that the construction of the national park has an impact on the livelihoods, main income, and non-agricultural income of the surrounding farmers.

**Table 1.** Overall descriptive statistics.

| Variable | | Description | 2015 | | 2022 | |
|---|---|---|---|---|---|---|
| | | | Mean | Std. Dev. | Mean | Std. Dev. |
| Farmers' livelihood choices | Engaged in agriculture | Yes = 1, No = 0 | 0.47 | 0.50 | 0.19 | 0.37 |
| Main income of farmers' households | Source of income | 0 = Agricultural income, 1 = Non-agricultural income | 0.31 | 0.46 | 0.63 | 0.48 |
| Covariate situation | Village leader | Yes = 1, No = 0 | 0.06 | 0.23 | 0.09 | 0.29 |
| | Party member | Yes = 1, No = 0 | 0.15 | 0.35 | 0.16 | 0.36 |
| | Family members | Actual Quantity | 5.10 | 2.48 | 5.02 | 2.07 |
| | Labor force quantity | Actual Quantity | 2.73 | 1.93 | 2.66 | 1.36 |
| | Annual total household savings | 1 = less than CNY 10,000; 2 = 1 to CNY 30,000; 3 = CNY 30,000 to 50,000; 4 = CNY 50,000 to 70,000; 5 = CNY 70,000 to 90,000; 6 = over CNY 90,000 | 2.04 | 1.62 | 2.42 | 1.77 |
| | Total planting area | Acre | 28.90 | 31.60 | 29.12 | 33.00 |
| | Tea plantation area | Acre | 19.64 | 25.69 | 20.54 | 25.82 |
| | Bamboo forest area | Acre | 6.66 | 18.45 | 6.79 | 19.18 |
| | Government compensation | Yes = 1, No = 0 | 0.28 | 0.45 | 0.46 | 0.50 |
| | Tea business income | CNY | 101,827.70 | 439,013.50 | 147,372.9 | 519,170.2 |
| | Attitude regarding work adjustment | Satisfaction level from low to high on a five-point scale | 2.48 | 1.15 | 3.48 | 1.05 |
| | Firm support efforts | Satisfaction level from low to high on a five-point scale | 1.68 | 1.00 | 2.05 | 1.21 |

**Table 1.** *Cont.*

| | Variable | Description | 2015 | | 2022 | |
|---|---|---|---|---|---|---|
| | | | **Mean** | **Std. Dev.** | **Mean** | **Std. Dev.** |
| | Huangcun | Yes = 1, No = 0 | 0.06 | 0.24 | 0.06 | 0.24 |
| | Hongxing Village | Yes = 1, No = 0 | 0.09 | 0.29 | 0.09 | 0.29 |
| | Xingcun | Yes = 1, No = 0 | 0.09 | 0.29 | 0.09 | 0.29 |
| | Chengdun Village | Yes = 1, No = 0 | 0.11 | 0.31 | 0.11 | 0.31 |
| Village-level | Caodun Village | Yes = 1, No = 0 | 0.09 | 0.29 | 0.09 | 0.29 |
| dummy variable | Zhoutou Village | Yes = 1, No = 0 | 0.11 | 0.31 | 0.11 | 0.31 |
| | Lixin Village | Yes = 1, No = 0 | 0.11 | 0.31 | 0.11 | 0.31 |
| | Nanyuanling Village | Yes = 1, No = 0 | 0.10 | 0.30 | 0.10 | 0.30 |
| | Huangbai Village | Yes = 1, No = 0 | 0.14 | 0.35 | 0.14 | 0.35 |
| | Daancun | Yes = 1, No = 0 | 0.10 | 0.30 | 0.10 | 0.30 |

Note: Engaged in agriculture: indicates that the surveyed farmers are engaged in agriculture or other industries. Source of income: indicates whether the main source of income for the surveyed farmers' households comes from agriculture or non-agricultural industries. Village leader: Is there anyone from the surveyed farmer's household serving as a village leader. Party member: are there any members of the Communist Party of China in the interviewed farmers' homes. Family members: number of surveyed farmers' family members. Labor force quantity: number of laborers in surveyed farmers' households. Annual total household saving: indicates the per capita annual savings amount of the surveyed farmers' households. Total planting area: indicates the total planting area of the surveyed farmers' households. Tea area: indicates the tea planting area of the surveyed farmers' households. Bamboo forest area: indicate the planting area of bamboo in the surveyed farmers' households. Government compensation: indicates whether the surveyed farmers' families received government subsidies that year. Tea business income: indicates the annual tea business income of the surveyed farmers' families. Attitude regarding work adjustment: expresses the attitude of the surveyed farmers' families towards their job adjustment. Firm support efforts: indicates the support provided by the township where the surveyed farmers are located for their employment. Village level dummy variable: indicates the situation of the research team in the research village.

### 3.2. The Impact of National Park Construction on Farmers' Livelihood Choices

Table 2 mainly analyzes the benchmark regression of the impact of the Mount Wuyi National Park construction on farmers' livelihood choices. As shown in the table: under the premise of controlling for village-level dummy variables, (1) represents univariate regression, (2) represents regression with partial control variables added, and (3) represents the regression with all control variables added. From the regression results, when the control variables are not considered, the construction of national parks has an insignificant positive effect on farmers' preference for agricultural choices. When the individual variables, land use variables, and attitude variables are added, there is a positive impact on the farmers' livelihood choices that favor agriculture around the Mount Wuyi National Park. Then, when the household variable is further increased, there is an insignificant result, that is, the construction of the Mount Wuyi National Park has an insignificant positive impact on the farmers' livelihood preference for agriculture. Therefore, from the overall results, the assumption H1 is valid.

### 3.3. The Impact of National Park Construction on the Main Sources of Income for Farmers

According to the analysis results in Table 3, the benchmark regression results of the impact of the construction of the Mount Wuyi National Park on the main sources of farmers' income can be obtained. As shown in Equation (1), when the control variables are not considered, the construction of national parks will have an insignificant negative impact on the main source of the income of farmers, that is, if they do not take advantage of the change in the main income of farmers' households and move toward non-agricultural income. In Formula (2), when considering all control variables except for the tea planting area, the construction of national parks has a significant negative impact on the main source of income for farmers. Equation (3) indicates that when considering all control variables, the construction of national parks will have an insignificant negative impact on the main source of income for farmers. Therefore, overall, assuming H2 is not established, the main

source of income for farmers in the process of national park construction still tends to be dominated by agricultural income.

**Table 2.** Benchmark regression analysis of the impact of national park construction on farmers' livelihood choices.

| Variables | Whether Engaged in Agriculture | | |
|---|---|---|---|
| | **(1)** | **(2)** | **(3)** |
| DID | 0.041 | 0.088 * | 0.062 |
| | (0.053) | (0.053) | (0.053) |
| Village leader | | 0.034 | 0.039 |
| | | (0.081) | (0.080) |
| Party member | | | −0.029 |
| | | | (0.097) |
| Family members | | | 0.003 |
| | | | (0.019) |
| Labor force quantity | | | 0.019 |
| | | | (0.024) |
| Annual total household savings | | −0.039 | −0.035 |
| | | (0.028) | (0.028) |
| Total planting area | | −0.011 | 0.010 |
| | | (0.013) | (0.015) |
| Tea area | | | −0.026 ** |
| | | | (0.013) |
| Bamboo forest area | | 0.015 | −0.005 |
| | | (0.019) | (0.021) |
| Government compensation | | −0.151 ** | −0.125 ** |
| | | (0.061) | (0.062) |
| Tea business income | | −0.000 | −0.000 |
| | | (0.000) | (0.000) |
| Attitude regarding work adjustment | | −0.031 | −0.021 |
| | | (0.021) | (0.021) |
| Firm support efforts | | −0.045 | −0.013 |
| | | (0.038) | (0.040) |
| Village-level dummy variable | Controlled | Controlled | Controlled |
| Constant | 0.319 *** | 0.886 *** | 0.769 *** |
| | (0.017) | (0.275) | (0.275) |
| Observations | 708 | 708 | 708 |
| R-squared | 0.763 | 0.780 | 0.784 |

Note: ***, **, and * represent significance levels of 1%, 5%, and 10%, respectively. DID: represents the core explanatory variable, which is the interaction term between the dummy variable of farmers grouping and the dummy variable of the establishment time of national parks. Engage in agriculture: indicates that the surveyed farmers are engaged in agriculture or other industries. Source of income: indicates whether the main source of income for the surveyed farmers' households comes from agriculture or non-agricultural industries. Village leaders: is there anyone from the surveyed farmer's household serving as a village leader. Party member: are there any members of the Communist Party of China in the interviewed farmers' homes. Family members: number of surveyed farmers' family members. Labor force quantity: number of laborers in surveyed farmers' households. Annual total household savings: indicates the per capita annual savings amount of the surveyed farmers' households. Total planting area: indicates the total planting area of the surveyed farmers' households. Tea area: indicates the tea planting area of the surveyed farmers' households. Bamboo forest area: indicates the planting area of bamboo in the surveyed farmers' households. Government compensation: indicates whether the surveyed farmers' families received government subsidies that year. Tea business income: indicates the annual tea business income of the surveyed farmers' families. Attitude regarding work adjustment: expresses the attitude of the surveyed farmers' families towards their job adjustment. Firm support efforts: indicates the support provided by the township where the surveyed farmers are located for their employment. Village-level dummy variable: indicates the situation of the research team in the research village.

**Table 3.** Benchmark regression analysis of the impact of national park construction on the main sources of income for farmers.

| Variables | Main Source of Household Income | | |
| --- | --- | --- | --- |
| | **(1)** | **(2)** | **(3)** |
| DID | −0.030 | −0.100 * | −0.084 |
| | (0.056) | (0.054) | (0.055) |
| Village leaders | | −0.007 | −0.003 |
| | | (0.076) | (0.077) |
| Party member | | −0.066 | −0.067 |
| | | (0.096) | (0.095) |
| Family members | | 0.015 | 0.013 |
| | | (0.019) | (0.019) |
| Labor force quantity | | −0.029 | −0.028 |
| | | (0.024) | (0.024) |
| Annual total household savings | | 0.080 *** | 0.077 ** |
| | | (0.030) | (0.030) |
| Total planting area | | 0.025 ** | 0.004 |
| | | (0.011) | (0.015) |
| Tea area | | | 0.026 ** |
| | | | (0.012) |
| Bamboo forest area | | −0.037 ** | −0.016 |
| | | (0.017) | (0.019) |
| Government compensation | | 0.146 ** | 0.119 * |
| | | (0.063) | (0.065) |
| Tea business income | | 0.000 * | 0.000 * |
| | | (0.000) | (0.000) |
| Attitude regarding work adjustment | | 0.044 ** | 0.035 |
| | | (0.022) | (0.023) |
| Firm support efforts | | −0.011 | −0.036 |
| | | (0.036) | (0.036) |
| Village-level dummy variable | Controlled | Controlled | Controlled |
| Constant | 0.480 *** | −0.383 | −0.334 |
| | (0.018) | (0.263) | (0.264) |
| Observations | 708 | 708 | 708 |
| R-squared | 0.762 | 0.787 | 0.789 |

Note: ***, **, and * represent significance levels of 1%, 5%, and 10%, respectively. DID: represents the core explanatory variable, which is the interaction term between the dummy variable of farmers grouping and the dummy variable of the establishment time of national parks. Engage in agriculture: indicates that the surveyed farmers are engaged in agriculture or other industries. Source of income: indicates whether the main source of income for the surveyed farmers' households comes from agriculture or non-agricultural industries. Village leader: is there anyone from the surveyed farmer's household serving as a village leader. Party member: are there any members of the Communist Party of China in the interviewed farmers' homes. Family members: Number of surveyed farmers' family members. Labor force quantity: number of laborers in surveyed farmers' households. Annual total household savings: indicates the per capita annual savings amount of the surveyed farmers' households. Total planting area: indicates the total planting area of the surveyed farmers' households. Tea area: indicates the tea planting area of the surveyed farmers' households. Bamboo forest area: indicates the planting area of bamboo in the surveyed farmers' households. Government compensation: indicates whether the surveyed farmers' families received government subsidies that year. Tea business income: indicates the annual tea business income of the surveyed farmers' families. Attitude regarding work adjustment: expresses the attitude of the surveyed farmers' families towards job adjustment. Firm support efforts: indicates the support provided by the township where the surveyed farmers are located for their employment. Village-level dummy variable: indicates the situation of the research team in the research village.

## 4. Discussion

### 4.1. Heterogeneity Analysis

Based on the on-site research conducted by the research group, this article categorizes the surveyed farmers by their special crop planting area (based on the research results, the tea planting area of local farmers is selected) and their household income, in order to analyze the heterogeneity of the impact of the construction of the Mount Wuyi National Park on farmers' livelihood choices. Of these, the tea planting area of farmers is divided

into a high planting area ($\geq$20 acres) and a low planting area (<20 acres). The household income situation is divided into high income (annual household income $\geq$ CNY 90,000) and low income (annual household income < CNY 90,000). According to the regression results in Column (1) and Column (2) of Table 4, the construction of the Mount Wuyi National Park significantly increased the probability of farmers in a high tea planting area regarding their livelihood choice in favor of agriculture by 15.7%, while the probability of farmers with low tea planting area regarding their livelihood choice in favor of agriculture significantly increased by 12.7%. The possible reason for this situation is that with the construction of the Mount Wuyi National Park, there are more and more restrictions on farmers' livelihood choices. For farmers who own a high tea planting area, it is more difficult to change their livelihood, so they will be more willing to focus on agriculture than farmers with a small tea planting area. According to the regression results in Column (3) and Column (4) of Table 4, from the perspective of income grouping, the construction of the Mount Wuyi National Park has significantly increased the probability of high-income farmers focusing on agricultural livelihood choices by 15.3%, while it has had no significant impact on low-income farmers. The main reason for this may be that the research team found that the main livelihood of farmers around the Mount Wuyi National Park is tea planting, bamboo economy, bamboo rafting, and tour guides. Of these, tea cultivation or management is the most common. In recent years, under the influence of factors such as COVID-19 and trade protection, the tea market in Mount Wuyi has declined compared with the past but still represents high income [35]. Therefore, from the perspective of the difficulty of industrial structure adjustment, the difficulty of farmers' own livelihood transformations and the limitations regarding the process of national park construction, surrounding farmers are willing to continue to maintain agricultural or related part-time activities while maintaining high incomes. For low-income farmers, after research, the research team found that due to the local environment and the disadvantages of farmers' own resource endowments compared to high-income families, most of them still maintain their original livelihood choices, and the impact of national park construction is not significant [36]. From this, it can be seen that Hypothesis 1 is proven and Hypothesis 2 is overturned.

**Table 4.** Heterogeneity analysis results.

| Variable | Explanatory Variable: Farmers' Livelihood Choices | | | |
| --- | --- | --- | --- | --- |
| | (1) | (2) | (3) | (4) |
| | High tea planting area | Low tea planting area | High Income | Low income |
| DID | 0.157 * | 0.127 ** | 0.153 * | 0.101 |
| | (0.094) | (0.059) | (0.084) | (0.061) |
| Control variable | Controlled | Controlled | Controlled | Controlled |
| Village-level dummy variable | Controlled | Controlled | Controlled | Controlled |
| Constant | 0.874 | 0.597 ** | 0.841 | 1.000 *** |
| | (1.021) | (0.265) | (0.708) | (0.317) |
| Observations | 260 | 448 | 351 | 357 |
| R-squared | 0.741 | 0.820 | 0.710 | 0.867 |

Note: ***, **, and * represent significance levels of 1%, 5%, and 10%, respectively.

### 4.2. Common Trend Testing

When using the DID model to analyze the impact of the construction of the Mount Wuyi National Park on the livelihood of surrounding farmers and the main incomes of farmers, it is necessary to ensure that relevant covariates meet the standard of the common trend test, that is, the treatment group and control group must meet the same trend of change before policy intervention. Otherwise, it is estimated that there will be deviations in the results [37]. Specifically, this paper refers to the research methods of Zhu Zhen et al. [38] to determine whether there are significant differences in the basic characteristics of farmers within and outside the National Park before the construction of the Mount Wuyi National

Park; if not, this indicates that the common trend hypothesis is reasonable and passes the test. Otherwise, it does not pass the common trend test.

As seen in Table 5, this article provides a common trend test analysis. And descriptive statistics were obtained from the surveyed farmers within and outside the national park in 2015 and 2022. The method used was econometric descriptive analysis. Through the statistical analysis of individual characteristic variables, household variables, land use variables, non-agricultural income variables, and the attitude variables of household heads inside and outside national parks in 2015 and 2022, corresponding results were obtained. The results showed that there were no significant differences in the personal characteristics variables, family variables, and attitude variables of the household heads of surveyed farmers both inside and outside the national parks in 2015 and 2022. However, there were significant differences in the total planting area, tea planting area, and annual tea income among the land use variables of the surveyed farmers both inside and outside the national park in the two years but the overall direction of the differences was the same. From this, it can be seen that although there are differences in the non-agricultural income variables of farmers within and outside the national park area between the two years, they did not change due to the implementation of policies. And other variables have relatively small differences in comparison between different years and regions. Therefore, it can be concluded that the DID model in this study meets the requirements of common trend testing.

**Table 5.** DID common trend test results.

| Variable | Within the Scope of National Parks | | | | Outside the Scope of National Parks | | | |
| | 2015 | | 2022 | | 2105 | | 2022 | |
| | Mean | SD | Mean | SD | Mean | SD | Mean | SD |
|---|---|---|---|---|---|---|---|---|
| Village leader | 0.06 | 0.24 | 0.09 | 0.29 | 0.05 | 0.10 | 0.10 | 0.30 |
| Party member | 0.16 | 0.37 | 0.17 | 0.38 | 0.11 | 0.13 | 0.13 | 0.34 |
| Family members | 5.06 | 2.40 | 5.16 | 2.10 | 5.15 | 4.75 | 4.75 | 2.00 |
| Labor force quantity | 2.72 | 1.84 | 2.78 | 1.36 | 2.75 | 2.10 | 2.43 | 1.34 |
| Annual total household savings | 2.29 | 1.73 | 2.76 | 1.87 | 1.58 | 1.78 | 1.78 | 1.35 |
| Total planting area | 31.42 | 34.46 | 31.81 | 36.27 | 24.17 | 24.832 | 24.08 | 25.12 |
| Tea area | 22.23 | 29.74 | 23.03 | 30.01 | 14.76 | 15.85 | 15.85 | 14.07 |
| Bamboo forest area | 6.66 | 17.47 | 6.90 | 18.49 | 6.66 | 6.59 | 6.59 | 20.50 |
| Government compensation | 0.26 | 0.44 | 0.48 | 0.50 | 0.32 | 0.47 | 0.42 | 0.50 |
| Tea business income | 144,619.00 | 533,692.00 | 208,099.60 | 628,605.1 | 21,463.41 | 103,507.60 | 33,325.20 | 121,866.10 |
| Attitude regarding work adjustment | 2.48 | 1.16 | 3.52 | 1.08 | 2.46 | 3.42 | 3.42 | 0.98 |
| Firm support efforts | 1.77 | 1.06 | 2.13 | 1.25 | 1.50 | 1.89 | 1.89 | 1.10 |

Note: Engage in agriculture: indicates that the surveyed farmers are engaged in agriculture or other industries. Source of income: indicates whether the main source of income for the surveyed farmers' households comes from agriculture or non-agricultural industries. Village leader: is there anyone from the surveyed farmer's household serving as a village leader. Party member: are there any members of the Communist Party of China in the interviewed farmers' homes. Family members: number of surveyed farmers' family members. Labor force quantity: number of laborers in surveyed farmers' households. Annual total household savings: indicates the per capita annual savings amount of the surveyed farmers' households. Total planting area: indicates the total planting area of the surveyed farmers' households. Tea area: indicates the tea planting area of the surveyed farmers' households. Bamboo forest area: indicates the planting area of bamboo in the surveyed farmers' households. Government compensation: indicates whether the surveyed farmers' families received government subsidies that year. Tea business income: indicates the annual tea business income of the surveyed farmers' families. Attitude regarding work adjustment: express the attitude of the surveyed farmers' families towards their job adjustment. Firm support efforts: indicates the support provided by the township where the surveyed farmers are located for their employment.

### 4.3. Placebo Test

To verify that the impact generated in the previous text was not affected by other policies, this article used the placebo test method to analyze the benchmark regression

results based on existing research results [39] (Figure 3). The main idea is to randomly select 231 households (equivalent to the total number of treatment groups) from all 354 households surveyed as the "hypothetical treatment group" and generate a "hypothetical variable interaction term" (i.e., the virtual variable interaction mentioned earlier). By conducting 500 regression analyses, it was concluded that, after 500 hypothesis tests, the corresponding regression coefficient distribution was obtained [40]. Most estimates of k-density are around 0. However, the estimated value of the policy coefficient in the original benchmark regression is far from it, as an outlier. From the *p*-value perspective, a large number of estimated results are above 0.1, indicating that the assumed regression results are not significant. These results indicate that the previous research findings are less likely to be influenced by other policies, so the DID research in the previous section is reliable.

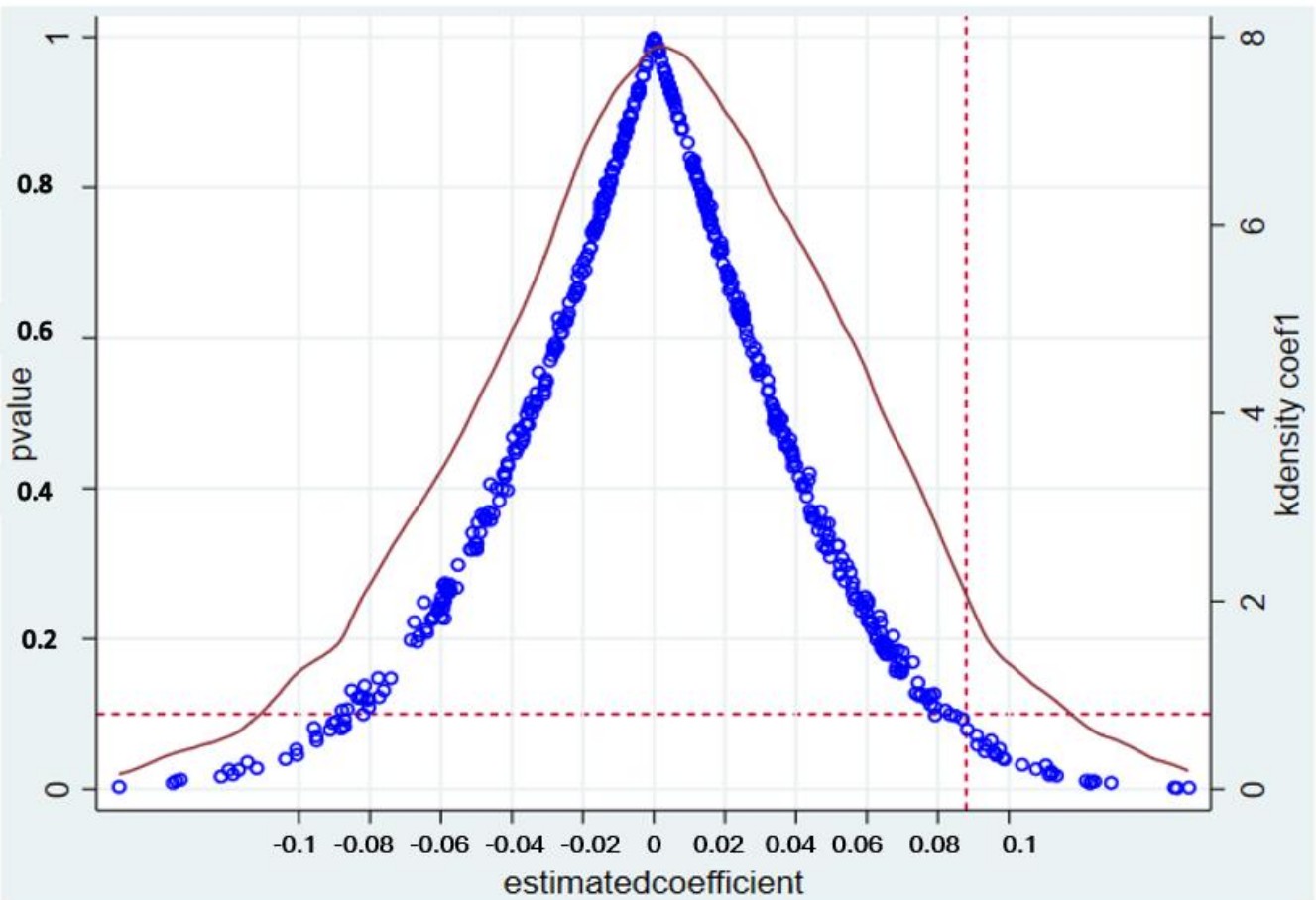

**Figure 3.** Placebo test results. Note: Red solid line: Represents the kernel density function image. Blue circle: Represents the *p*-value. The vertical red dashed line: Represents the true coefficient value, which is 0.088. The horizontal red dashed line: Represents the true *p*-value, which is 0.1.

## 5. Conclusions

Based on the establishment of the Mount Wuyi National Park as an example, this paper analyzes the impact of the establishment of national parks on the livelihood of surrounding farmers and the main sources of income of families. Meanwhile, this article analyzes the impact of different tea planting areas and household income on farmers' livelihood choices. The results show that: First, with the establishment of the Mount Wuyi National Park, the restrictions on farmers' economic activities have increased. Moreover, the price of tea within the national park is significantly higher than that outside the national park. Therefore, farmers within the scope of national parks are more inclined to maintain tea related agricultural or part-time livelihoods than those outside the scope of national

parks. Second, after the establishment of Mount Wuyi National Park, the main income of surrounding farmers' families has not changed to non-agricultural income, as suggested in the previous hypothesis, but the research results have a negative impact, which indicates that surrounding farmers still rely on agricultural-related income as the main source of family income. Thirdly, under different levels of tea cultivation and household income, there are also certain differences in the livelihood choices of farmers in different groups. After the construction of national parks, high tea planting areas and high-income farmers are more inclined to maintain the agricultural livelihood choice of mainly planting tea. Fourthly, the authors found through the field research and data analysis of the research group that the interviewed farmers generally reported that after the establishment of Mount Wuyi National Park, the restrictions on farmers in the park were strengthened, and the restrictions on farmers outside the park were reduced. As a result, farmers in the Mount Wuyi National Park tended to mainly be employed in tea industry. Farmers outside the park experienced a certain degree of livelihood transformation.

Through the analysis of the above research conclusions, we reached the following conclusions. First, with the establishment of the Mount Wuyi National Park, surrounding farmers are still subject to the management of national parks, resource endowment conditions and other factors, and their livelihood choices are still dominated by tea planting. However, during the research process, it was found that many tea farmers reported too many restrictions that affected their returns. Therefore, in the management process of the Mount Wuyi National Park, it is necessary to further improve relevant laws and regulations on the premise of ecological protection and moderately relax farmers' reasonable livelihood choices. Secondly, in previous research, it was found that government subsidies were an important factor affecting farmers' livelihoods. However, during the research process, the research team found that, especially for farmers within the scope of national parks, there is a problem of the inadequate implementation of subsidies. Therefore, the subsidy work for surrounding farmers needs to be further strengthened in terms of supervision and implementation. Third, it was found in this study that farmers with low income and less tea planting area around the Mount Wuyi National Park have weak willingness to choose agricultural livelihoods. It is recommended that the government carry out professional livelihood training activities for farmers. Fourthly, based on the previous research and actual investigation, it was found that the income of farmers around the Mount Wuyi National Park who rely on tea to choose their livelihood is unstable. Therefore, it is suggested that during the construction process of national parks, the advantages of surrounding natural conditions should be utilized, and the government should further guide farmers to utilize local tourism and cultural industry resources in order to diversify their livelihood choices and thus gradually increase their incomes.

**Author Contributions:** Conceptualization, Z.Y. and J.R. methodology, Z.Y.; formal analysis, Z.Y. and J.R.; investigation, Z.Y.; resources, D.Z.; data curation, Z.Y.; writing—original draft preparation, Z.Y.; writing—review and editing, J.R.; supervision, D.Z.; project administration, D.Z.; funding acquisition, D.Z. All authors have read and agreed to the published version of the manuscript.

**Funding:** This research was funded by the 2014 National Forestry and Grassland Administration Forestry Major Issues Seminar Project (ZDWT201415).

**Institutional Review Board Statement:** This study is not related to ethical issues.

**Data Availability Statement:** Not applicable.

**Conflicts of Interest:** The authors declare no conflict of interest.

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
