# Peer review of "The Impact of the Establishment of the Mount Wuyi National Park on the Livelihood of Farmers"

_agriculture, doi:10.3390/agriculture13081619_

Round 1

Reviewer 1 Report

Some comments and suggestions on the research paper for agriculture-2531627

At first, thank you for your invaluable research about the Impact of Mount Wuyi National Park Construction on Farmers' Livelihoods and Main Income.

I have some comments and suggestions on this manuscript.

Title:

Not need to word “research on”

Abstract:

Edit the sentence on line 14-15 and it is without verb.

Improve the results section in the abstract and also conclusion.

Main income is not a good keyword, please change it.

Introduction:

I think construction is not a good definition for national parks, revise it.

What is CPC in line 28, define it for the first time and then use its acronym.

In line 53, some scholar, which one? State at least 2 or 3 references.

Check all the text in English, there are a lot of mistakes. Line 216-217, lines 254-255.

One–on–one or face-to-face, check it, please.

Edit the first phrase of the paragraph, lines 236-247-256. Line 439-440.

Check the figure numbers. Figure 1 for two figures.

Please add the full name of the variable in the bottom of each table for more clarification.

In table 5, there is needed a statistical analysis between two years in within the scope of national parks and outside the scope of national parks. Also, add in the method section.

Please draw figure 3 with high quality. The number in the x-axis is messed up.

Check all the text in English, there are a lot of mistakes. Line 216-217, lines 254-255.

Author Response

Dear reviewer,

Hello! I am the first author of the article. Thank you for your suggestions on the revisions to our article. I have now completed the modifications according to your requirements. The specific modifications are as follows: Firstly, regarding the title of the article, I have removed the word "research" and replaced the word "construction" with "establishment". Secondly, I have improved the sentences on lines 14 to 15 of the article. Thirdly, I have made modifications to the results section of the abstract and conclusion. Fourthly, I have made modifications to the 'main income'. Fifth, I have changed the construction of national parks to "National Park Establishment". And replaced the words "construction" in the article with "establishment". Sixth, in line 28 of the article, I replaced "CPC" with a member of the Communist Party of China. Seventh, in line 53 of the article, I replaced it with two or more scholar names. Eighth, the text in lines 216 to 217 and 254 to 255 of this chapter is incorrect, and I have also made modifications. I have made modifications to the first sentence of lines 236, 247, 256, 439, and 440. Tenth, regarding Figure 1 appearing twice, I have also made modifications. Eleventh, at the bottom of each table, I added the full names of variables and provided explanations. Twelfth, in accordance with your opinion, I have added statistical analysis and methodological analysis within and outside the scope of the national park and two time periods. Thirteen, regarding Figure 3, I have made corrections. Fourthly, regarding the PDF version you provided, I have also made modifications according to the parts highlighted in yellow inside.

The above are the modifications I have made to your feedback. I have also made modifications to the text. The part I modified is the highlighted text in red in the submitted Word. Thank you again for your thoughtful guidance, teacher. We look forward to your feedback on the next steps as soon as possible.

Yours Sincerely,

Yang Zhen

Reviewer 2 Report

The present work makes an effort to estimate the impact of Mount Wuyi National Park construction on the livelihood choices of surrounding farmers and the 13 main income of families. The topic is interesting lies within the scope of the journal though amendments are necessary in order to become publishable.

The literature review should be encriched with publ more focused to the topic of the manuscript;

Tan, S., Zhong, Y., Yang, F., & Gong, X. (2021). The impact of Nanshan National Park concession policy on farmers' income in China. Global Ecology and Conservation31, e01804.

Desmiwati, D., Veriasa, T. O., Aminah, A., Safitri, A. D., Wisudayati, T. A., Hendarto, K. A., ... & Sari, D. R. (2021). Contribution of agroforestry systems to farmer income in state forest areas: A case study of Parungpanjang, Indonesia. Forest and Society5(1), 109-119.

In addition the conclusions should document more conceise and focused and better documented by the results obtaones through the analysis

more policy implicatios

The present work has major issues in english that makes in cases difficult to understand the meaning(ie abstrac 3d line, 1st paragraph conclusions etc

Author Response

Dear Editor,

Hello! I am Yang Zhen, the first author of the article. I am very grateful for your valuable feedback on our article. The following are my modifications to your feedback.

Firstly, I have re described the variables in detail. And at the bottom of the table, I also add abbreviations, full names, and explanations for variables. Secondly, I explained the selection of samples and included grouping. Thirdly, in the introduction section, I supplemented the main objectives and innovations of this study. Fourthly, in the conclusion of this article, I have added a supplementary section on policy implications.

The above are the modifications I have made to your feedback I have also made modifications to the text The part I modified is the highlighted text in red in the submitted Word Thank you again for your thorough guidance, teacher We look forward to your feedback on the next steps as soon as possible.

Yours Sincerely,

Yang Zhen

Reviewer 3 Report

1.        The title of the manuscript needs to be corrected, it is not problematic, it does not communicate what will be the subject of the research.

2.        The concept of ‘Mount Wuyi National Park Construction’ is central to the whole study. The term used (national park construction) is too literal. What exactly do the Authors mean (also taking into account the history of the park described in section 2.2), is the term ‘park construction’ synonymous with ‘Park was officially established’ e.g. line 191. This issue needs to be clarified. A question also arises as to what this phrase has to do with the research issues/problems discussed so far in the literature.

3.        The topic requires better and broader embedding in the literature, it is necessary to refer to the issue raised many times in the literature, i.e. the relationship between nature conservation and human well-being, the idea of sustainable development.

4.        Research intentions are vague, vaguely described (Lines 110-116)

5.        It is incomprehensible to me to formulate research intentions in the future (research questions 116-120; hypotheses), when the surveys concerned the period 2015-2022.

6.        There is no clear indication of the time range of the research (not the time of the research), which makes it difficult to understand the preliminary considerations regarding the issue under consideration. I leave for consideration the earlier (than in section 2.2) placement of basic information about when the park was created, etc.

7.        In order to better understand the role of the park for the area, it is required to characterize the activities undertaken by the inhabitants of the surrounding villages around the park before its launch (maybe referring to the survey according to the state of 2015?) along with presenting the income situation of the inhabitants. I hope it will also explain and understand why the Authors dealt with farmers only.

8.        The text is incomprehensible (also linguistically) (lines 208-215). It was stated that the park is representative (line 215) - how to justify this/what is the reason for this?

9.        A weak point is the discussion of research results with studies of other authors. Line 462 –in the previous research’- what research is it about?

10.     The lack of indication of the limitations of the study is a source of dissatisfaction.

11. Linguistically, the whole manuscript definitely needs linguistic improvement - often sentences are without a subject, there are sentence equivalents.

Author Response

Dear Editor,

Hello! I am Yang Zhen, the first author of this article. Thank you, teacher, for providing us with so many valuable opinions. I have made individual modifications to your suggestion. Here are my modifications.

Firstly, I have made modifications to the title of the article. It is changed to "the impact of the establishment of Mount Wuyi National Park on the livelihood of farmers". Secondly, in your second suggestion, I made a modification on line 191 of this chapter. And I believe that "establishment of national parks" will be closer to the theme of the article than "construction of national parks". The establishment of Mount Wuyi National Park is actually the core of this study. Due to the establishment process of national parks, it has had an impact on the livelihoods of surrounding farmers and the main income of their families. Thirdly, I have included sustainable development and harmonious coexistence between humans and nature in this chapter. Fourthly, I have made modifications to lines 110-116 of this chapter. Fifthly, the research question of this paper is: by analyzing how the establishment of Mount Wuyi National Park affected the livelihood choices of surrounding farmers and the main income of families. The reason for choosing 2015 and 2022 is that 2015 is the year before Mount Wuyi National Park becomes a pilot project. 2022 is the year after the official establishment of Mount Wuyi National Park. This can easily form a contrast. The assumption in this article is mainly about the impact of farmers' livelihood choices and their main household income. Sixth, I added the reasons for choosing the research time frame in the article. Seventh, I added activities of surrounding residents before the establishment of Mount Wuyi National Park in the article. Eighth, I made modifications to lines 208-215 of the article and made modifications to the "representativeness" section. Ninth, I have also made revisions to line 462 of the article. Tenth, I have supplemented the limitations of the article's research. Eleven, regarding the language errors in the article, I have also made revisions.

The above are the modifications I have made to your feedback I have also made modifications to the text The part I modified is the highlighted text in red in the submitted Word Thank you again for your thorough guidance, teacher We look forward to your feedback on the next steps as soon as possible

Yours Sincerely,

Yang Zhen

Round 2

Reviewer 2 Report

All the comments have been addressed. It is publishable

The English is satisfactory